# Process and Techno-Economic Analysis for Fuel and Chemical Production by Hydrodeoxygenation of Bio-Oil

Giuseppe Bagnato  and Aimaro Sanna *

Advanced Biofuels Lab, Institute of Mechanical, Process and Energy Engineering (IMPEE), School of Engineering & Physical Sciences, Heriot-Watt University, Edinburgh EH14 4AS, UK; gb17@hw.ac.uk
* Correspondence: a.sanna@hw.ac.uk

**Abstract:** The catalytic hydrogenation of lignocellulosic derived bio-oil was assessed from the thermodynamic simulation perspective, in order to evaluate its economic potential for the production of added-value chemicals and drop-in fuels. A preliminary economic evaluation was first run to identify the conditions where the process is profitable, while a full economic analysis evaluated how the operating conditions affected the reaction in terms of yield. The results indicate that the bio-oil should be separated into water-soluble and insoluble fractions previous hydrogenation, since very different process conditions are required for the two portions. The maximum economic potential resulted in 38,234 MM$/y for a capacity of bio-oil processed of 10 Mt/y. In the simulated biorefinery, the insoluble bio-oil fraction (IBO) was processed to produce biofuels with a cost of 22.22 and 18.87 $/GJ for light gasoline and diesel, respectively. The water-soluble bio-oil fraction (WBO) was instead processed to produce 51.43 ton/day of chemicals, such as sorbitol, propanediol, butanediol, etc., for a value equal to the market price. The economic feasibility of the biorefinery resulted in a return of investment (ROI) of 69.18%, a pay-out time of 2.48 years and a discounted cash flow rate of return (DCFROR) of 19.11%, considering a plant cycle life of 30 years.

**Keywords:** bio-oil; catalysis; hydrogen; bio-fuels; process design

## 1. Introduction

In 2018, fossil fuels' share in global energy production was 136,580 TWh (93.6% of the total) [1] contributing to the increase in greenhouse gases (GHG) in the atmosphere, which are gradually raising the global temperature, thus causing a series of problems to our planet [2,3]. Currently, the scientific community, in sync with national and international policies (e.g., COP21), are seeking into alternative source of clean energy for reducing GHG emissions. Renewable energy represents energy derived from renewable sources such as solar energy, wind power, hydroelectric power, geothermal energy, tidal power and biomass [4]. Among them, biomass is the only renewable source that can cover all three aspects of energy uses: electricity, heat and transportation fuels. In particular, its densification in liquid bio-oils through thermochemical processes such as pyrolysis and the further upgrading of the oils using crude oil refineries is attractive due to the possibility to carry on using existing infrastructures. The bio-oils obtained from lignocellulosic biomass are dark brown organic liquids with the presence of many different organic compounds such as aldehydes, ketones, sugars, carboxylic acids and phenols. However, the potential of these liquids for the direct substitution of petroleum fuels is limited due to their high viscosity, high water and oxygen contents, low heating value, instability and high acidity (corrosiveness).

The catalytic hydrodeoxygenation (HDO) of biomass-derived fast pyrolysis oil represents a fascinating route for the production of liquid transportation fuels and commodity chemicals. The path



for the conversion of biomass into the petroleum-compatible product through pyrolysis/HDO can be divided into a series of steps including feed purification, chemical modification and products separation. In refineries, hydrogenation reactions are common operations used to limit the presence of oxygen, nitrogen, sulphur, olefins and aromatics [5]. The reaction is catalysed by molybdenum together with Ni or Co supported by $\gamma Al_2O_3$. The operating conditions depend on the type of feed: Liquid hourly space velocity (LHSV) ranges from 0.2 to 8.0, $H_2$ flow from 50 to 675 $Nm^3/m^3$, $H_2$ pressure between 14 and 138 bar and temperatures between 290 and 470 °C [6].

According to the feed processed and the desired products, different metals can be used. The metals reactivity scale:

$$\text{Olefin: Rh, Ru, Pt, Pd} > \text{Ni, Ir} > \text{Co, Fe,} \tag{1}$$

$$\text{Aromatic: Pt} > \text{Rh, Ru} > \text{Pd, Ni} > \text{Co, Fe.} \tag{2}$$

Noble metals are able to hydrogenate olefins and aromatics compound compared to conventional metals. Acid support is present, favouring the isomerisation reactions but also promoting coke formation. The acid support reactivity decreases as follow:

$$\text{Zeolite} > SiO_2/Al_2O_3 > H_3PO_4/SiO_2 > Al_2O_3. \tag{3}$$

Noble catalysts supported on C (Ru/C, Ru/TiO_2, Ru/Al_2O_3, Pt/C, and Pd/C) have been studied for the hydrotreatment of bio-oil by a number of authors. For example, Wildschut et al. [7] studied the bio-oil hydrogenation at different temperatures (250 and 350 °C) and pressures (100 and 200 bar) and compared carbon-supported catalysts with conventional hydrotreatment catalysts (sulfided NiMo/Al_2O_3 and CoMo/Al_2O_3). The authors obtained best performance with Ru/C catalyst in term of oil yield (up to 60 wt %) and deoxygenation level (up to 90 wt %). Ardiyanti et al. [8] tested different noble mono and bimetallic catalyst (Pt, Pd, Rh) supported on zirconia at 350 °C and at 200 bar. The yields of the upgraded bio-oils resulted in 37 and 47 wt % oil (based on the feed), the remainder being an aqueous phase (30–42 wt % based on feed), a gas phase (6–10 wt % on feed) and some coke (2–7 wt % on feed). Furthermore, the noble catalyst showed higher activity than CoMo/Al_2O_3 under the same condition. Furthermore, Ardiyanti et al. [9] proposed a reactivity scale for the bimetallic catalysts for bio-oil HDO, as follow:

$$Pd/ZrO_2 > Rh/ZrO_2 > RhPd/ZrO_2 \approx PdPt/ZrO_2 > RhPt/ZrO_2 > Pt/ZrO_2 > CoMo/Al_2O_3. \tag{4}$$

Pucher et al. [10] tested the performance of noble (Ru, Pt and Pd) and Ni catalysts at moderate (250 °C and 100 bar) and severe (at 300 °C and 150 bar) conditions in a bath reactor. Pt/C showed good results in term of increasing the calorific value of the upgraded bio-oils and reduction of coke formation; also, the water content was reduced, as about 86% and 73% to 79% of the starting energy was transferred into the oil phase. Instead, the use of Ru/C catalysts resulted in a higher $H_2$ consumption up to about 200 °C. After this temperature, the polymerization reactions were favoured and the $H_2$ consumption remained constant [11]. NiMo/$\gamma$-Al_2O_3 has been studied [12] at 390 °C and 70 bar. In this study, the bio-oil was heated inside the reactor injecting hot hydrogen, resulting in a decrement of coke and the formation of aromatic ring compounds. To reduce the acidity value of bio-oil, Parapati et al. [13,14] added 0.5 mg KOH/g bio-oil, carrying out the bio-oil hydrotreating with sulfided CoMo/$\gamma$-Al_2O_3 and KOH treated reduced CoMo/$\gamma$-Al_2O_3 (2 mg KOH/g bio-oil) at 375 °C and 100 bar. In each test, they obtained an oxygen content around 0.1%, and the higher heating value (HHV) increased to about 44 MJ/kg. Moreover, the products of the reduced catalyst resulted in 50% gasoline, 30% jet fuel and 20% diesel, while the sulfided catalyst produced 90% gasoline, 5% jet fuel and 5% diesel. The bio-fuel obtained in literature have been presented as Van Krevelen plot in Figure 1, where the bio-fuel proprieties in term of O/C and H/C molar ratios are compared. A bio-fuel with the same proprieties of the diesel should have O/C and H/C molar ratios that tend to zero and are greater than

one, respectively. The bio-fuels obtained by Parapati et al. [13,14] resulted comparable to diesel, thanks to the strong deoxygenation obtained using the reduced catalysts in presence of a strong base.

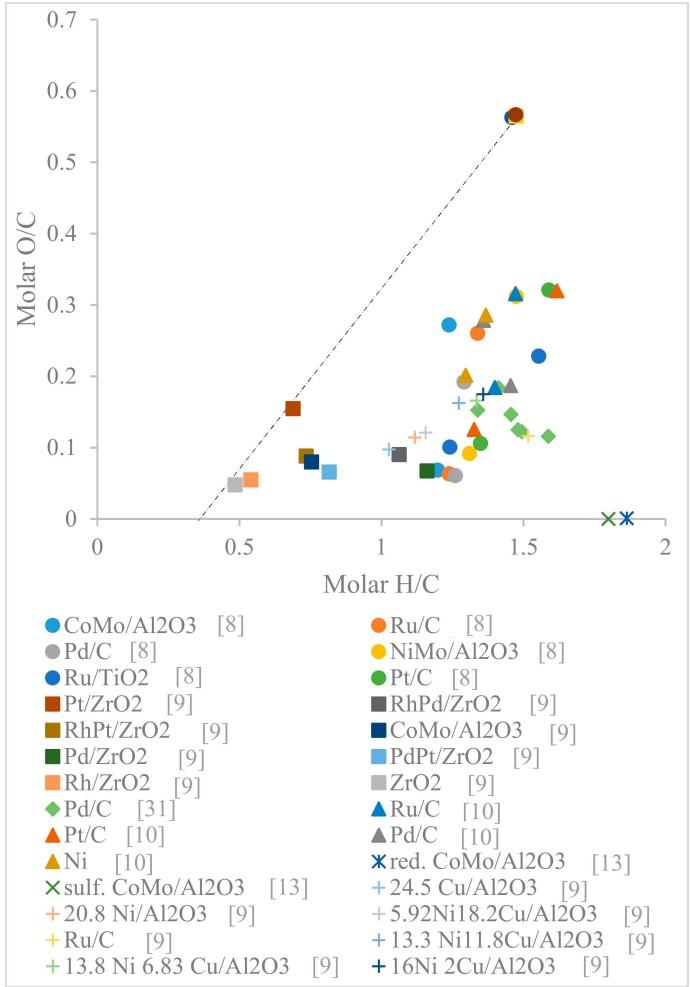

**Figure 1.** Van Krevelen plot.

Recent works analysed the techno-economics of hydrotreating bio-oil to biofuels [15–19]. Bagnato et al. [15] simulated the pyrolysis of 2000 dry ton/day of *Isochrysis* sp. Microalgae in the presence of Li-LSX-zeolite, producing biogas, bio-oil and bio-char. Subsequently, the bio-oil was upgraded by a hydrotreating reaction into fuels, achieving a minimum fuel selling price of 1.418 $/L. Wright et al. [17] obtained a minimum fuel selling price (MFSP) of $2.48 per gallon from the bio-oil chematizedn and hydrotreating in a 1440 ton/day plant and showed that hydrogen from bio-oil reforming resulted in the lowest biofuel emissions, but is not always economical. Carrasco et al. [18] studied the TEA of converting forest residues by pyrolysis and the further bio-oil upgrade by hydro-treatment and simultaneous production of $H_2$ for a feed rate of 2000 dry ton/day, obtaining a MFSP of $1.27 per litre. The main economic concerns were linked to high CAPEX and feedstock cost and short hydrotreating catalyst lifetime. Finally, Zhu et al. [19] designed a process for the high thermal liquefaction of 2000 dry metric ton/day wood biomass derivate, estimating a MFSP equal to 0.98 $/L-equivalent. The studies above show high variability of the MFSP and are mostly focused on producing transport fuels, while little is reported on producing added-value chemicals from the bio-oil. Therefore, this study aims to evaluate the overall performance of a bio-oil hydrotreating process to both transportation fuels and chemicals and explore the appropriate operation variables by the economic criteria suggested by Douglas et al. [20]. This work is performed using Aspen Plus software based

on the simulation of the hydrogenation reactions of the water soluble and insoluble bio-oil process system, with a feedstock processing capacity of 10 Mt/y.

## 2. Results and Discussion

The designed simulated process for the HDO of biomass-derived fast pyrolysis oil is chematized in Figures 2 and 3 for WBO and IBO fraction, respectively. For both fractions there is a "preparation zone", where the reactants are heated up and the gas streams are compressed; a "reaction zone" where the reactants are converted into the desired products. This zone consists to a series of adiabatic reactors with intermediate cooling system. The last part of the plant is the "separation zone", which is divided into a vapour and liquid recovery for the WBO and a distillation tower for the IBO.

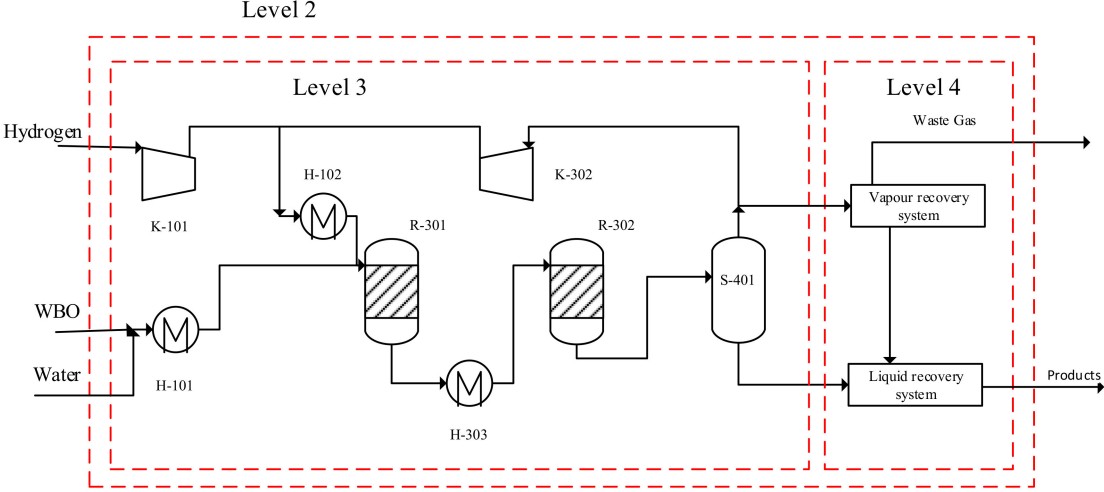

**Figure 2.** Hydrodeoxygenation process for the water soluble bio-oil.

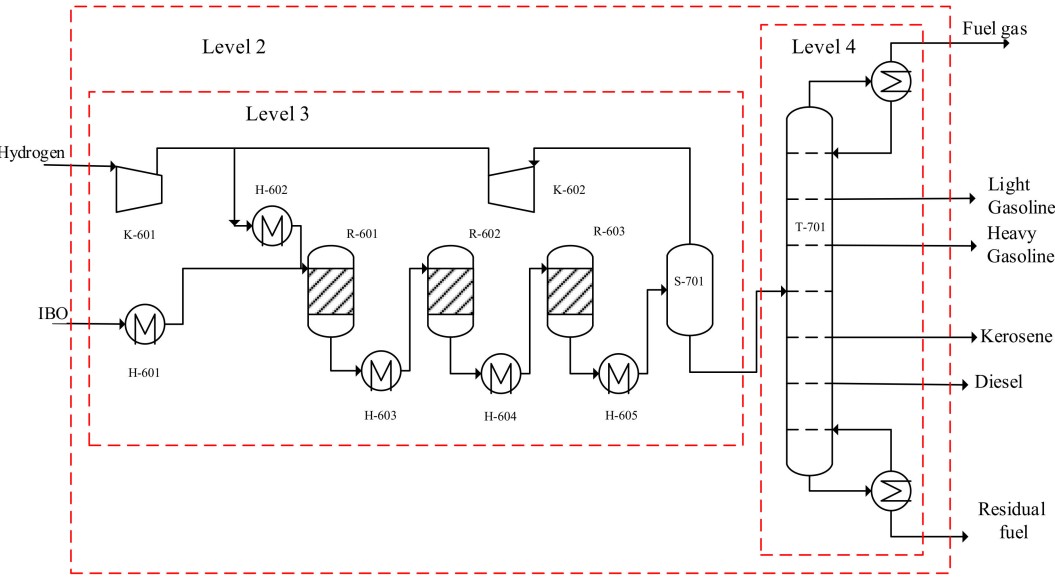

**Figure 3.** Hydrodeoxygenation process for the insoluble bio-oil.

### 2.1. Level 0: Preliminary Information

The bio-oil composition of the feed considered in this work was simplified using only the most representative compounds of the different pinewood bio-oil functionalities (see Table 1). Additionally, it was assumed that (1) the fresh feed does not contain impurities such as ash and solid particles;

(2) the bio-oil is separated in two phases, a water-soluble bio-oil (WBO) and a water-insoluble bio-oil (IBO).

**Table 1.** Bio-oil composition.

| Compound Group | Model Compound | Formula | wt % Dry Base | Ref. |
|---|---|---|---|---|
| Water-soluble | | | | |
| Acids | Acetic acid | $C_2H_4O_2$ | 3.90 | [21] |
| | Levulinic acid | $C_5H_8O_3$ | 1.67 | |
| Alcohols | Tyrosol | $C_8H_{10}O_2$ | 3.48 | [21–23] |
| | Glycerol | $C_3H_8O_3$ | 3.49 | |
| Ketones | Hydroxyacetone | $C_3H_6O_2$ | 8.29 | [21–23] |
| Aldehydes | Hydroxyacetaldeyde 3-Methoxy-4-Hydroxybenzaldehyde | $C_8H_8O_3$ | 6.97 | [21,23–25] |
| Guaiacols | o-Methoxyphenol | $C_{10}H_{12}O_2$ | 4.98 | [21,22,26,27] |
| Low MW sugars | Levoglucosan | $C_6H_{10}O_5$ | 5.97 | [22,23,26,28] |
| High MW sugars | Cellobiose | $C_{12}H_{22}O_{11}$ | 33.86 | [26] |
| Water-insoluble | | | | |
| Low MW lignin-derived compounds | Dimethoxy stilbene | $C_{16}H_{16}O_2$ | 10.95 | [28] |
| | Dibenzofuran (representing diphenyl compounds) | $C_{12}H_8O$ | 2.21 | [29] |
| Extractives | Dehydroabietic acid | $C_{20}H_{28}O_2$ | 2.99 | [26–28] |
| High MW lignin-derived compounds | Oligomeric compounds with β-O-4 bond | $C_{20}H_{26}O_8$ | 9.15 | [30] |
| | Phenylcoumaran compounds | $C_{21}H_{26}O_8$ | 1.99 | [29] |
| Nitrogen compounds | 2,4,6-trimethylpyridine | $C_8H_{11}N$ | 0.070 | [31] |
| Sulfur compounds | Dibenzothiophene | $C_{12}H_8S$ | 0.025 | [23] |

To design the HDO process, experimental data from literature were taken into account. The reaction pathways considered for the WBO hydrogenation are represented in Figure 4, while the reaction mechanisms of the IBO fraction hydrogenation can be seen in Figure 5 [30]. For the WBO fraction, the equation kinetics for the hydrogenation reaction were identified in literature [32–37], using as catalyst 5 wt % Ru/C. Polymerization reactions that are favored at temperature higher than 200 °C, were also taken into consideration in the model [8].

For the IBO fraction, the kinetic equation of Yu-Hwa et al. was considered for the simulation [38], where the IBO was divided into six groups (heavy non volatiles, light non-volatile, phenols, aromatics, alkanes, coke + $H_2O$ + outlet gases) and CoMo/$\gamma$-$Al_2O_3$ was the selected catalyst.

To simplify the evolution of the data, the HDO products were separated in specific streams according to their temperature boiling point (TBP): (1) fuel gas stream (up to 35 °C); (2) (3) light gasoline (35–90 °C); (4) heavy gasoline—excluding water (90–180 °C); (5) kerosene (180–250 °C); (6) diesel (250–350 °C) and finally, (7) residual fuel (more than 350 °C).

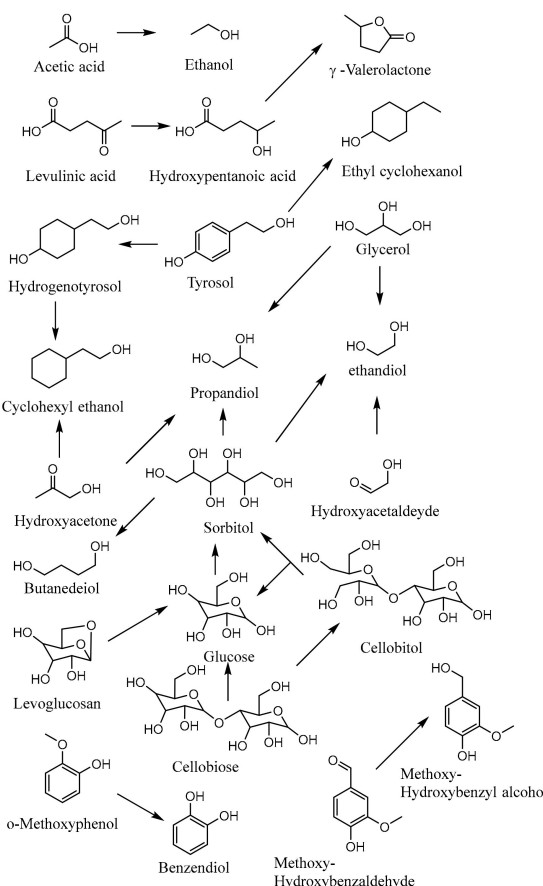

**Figure 4.** HDO reaction pathways for the water soluble bio-oil.

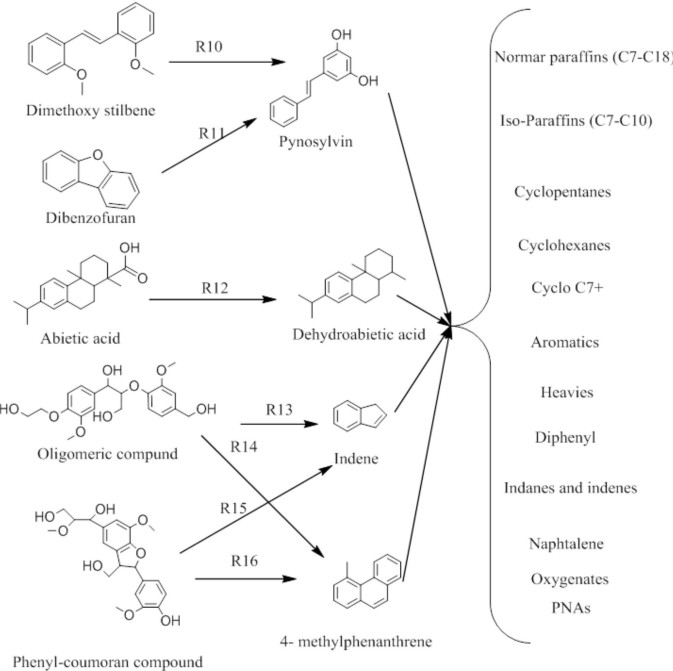

**Figure 5.** HDO reaction scheme for the insoluble bio-oil (modified from [30]).

## 2.2. Level 1: Batch versus Continuous

Continuous processes are designed to be operational 24 h per day, without interruption, contrary for batch process. The criterion to be used to choose a continuous or batch process depends on the plant capacity; if the plants have a capacity greater than 4500 ton/year, they are usually continuous [39]. In our case, the process capacity was assumed to be 10 million ton/year, which is similar in terms of raw material processed (215 thousands barrel per day) to the Valero Refining New Orleans LLC [40]. Therefore, the plant was simulated in continuous mode. Furthermore, the process was divided so that the (i) WBO fraction (representing 75% of the organic phase) was evaluated for the production of chemicals and fuels, and (ii) the IBO fraction (12.5% of the whole bio-oil, equal to 4300 m$^3$/day) for fuels production.

## 2.3. Level 2: Input-Output Structure

The HDO of the bio-oil was studied in terms of product yields, by varying the reaction temperature between 50 and 500 °C, at pressures from 10 to 150 bar and changing the H$_2$/bio-oil feed molar ratio from 1 to 4. Furthermore, the sorbitol yield was separately investigated, as the reaction is strongly thermodynamic limited.

Water bio-oil fraction: The HDO reaction for the WBO involves multiple reactions, of which, three of them with thermodynamic equilibrium. In particular:

$$Hydroxyacetone + H_2 \leftrightarrow Propanediol \ \Delta H < 0, \tag{5}$$

$$Cellobiose + H_2O \leftrightarrow 2 \ Dextrose \ \Delta H > 0, \tag{6}$$

$$Dextrose + H_2 \leftrightarrow Sorbitol \ \Delta H < 0. \tag{7}$$

Being the reactions exothermic, Equations (5) and (7) are favoured at low temperature and high pressure due and a negative variation of moles number ($\Delta v$) [41,42]. On the contrary, reaction (6) is favoured at high temperature, but the pressure is not of influence because the $\Delta v = 0$. One of the most important steps in the HDO of bio-oils is the conversion of levoglucosan and glucose to sorbitol, which represent the limiting step of the WBO–HDO process. From sorbitol, a number of shorten chain hydrogenation/hydrodeoxygenation products can be then obtained by varying the process conditions [43]. Therefore, a better understanding of the effect that the operating conditions have on the sorbitol yield is crucial to design a tuneable bio-oil HDO process. Figure 6 shows the yield variation of (a) sorbitol, (b) cellobiose and (c) in function of temperature, pressure and H$_2$/WBO molar ratio. The maximum sorbitol yield was obtained at the minimum temperature, maximum reaction pressure and feed molar ratio studied (50 °C, 150 bar and H$_2$/WBO = 4). Analysing Figure 6a–c, it can be seen that the yield variation is minimal when the H$_2$/WBO molar ratio is changed from 2 to 4 up to about 80 bar. Furthermore, the cellobiose yield is very low in all the cases analysed, where it is mostly converted in glucose. Analysing the glucose distribution (Figure 6a), a pressure higher than 40 bar and temperature up to 90 °C favoured the conversion of glucose in sorbitol, this is possible to evaluate from the sorbitol distribution. The difference in yield at the different feed molar ratios enlarges when the temperature is higher than 120 °C. To maximize the sorbitol yield, the cellobiose and glucose yields must be reduced, since their yield is inversely proportional to that of sorbitol. Previous work shows that 91.4% of glucose can be converted to sorbitol in presence of Ru-carbon nanotubes at 130 °C, 20 bar in aqueous phase with 98.2% selectivity [44]. Based on the thermodynamic calculations, in presence of a dimer as cellobiose, which must be hydrolysed before its hydrotreating, the formation of sorbitol is favoured at lower temperatures (<60 °C) and higher pressure (>40 bar).

Another work showed that hydrolysis of cellobiose does not occur in a neutral environment such as water and that the simultaneous hydrolysis (in ZnCl$_2$4H$_2$O) and hydrogenation (in presence of H$_2$ and Ru/C) is favoured at 125 °C and 40 bar, since at lower temperatures there is no hydrolysis [45].

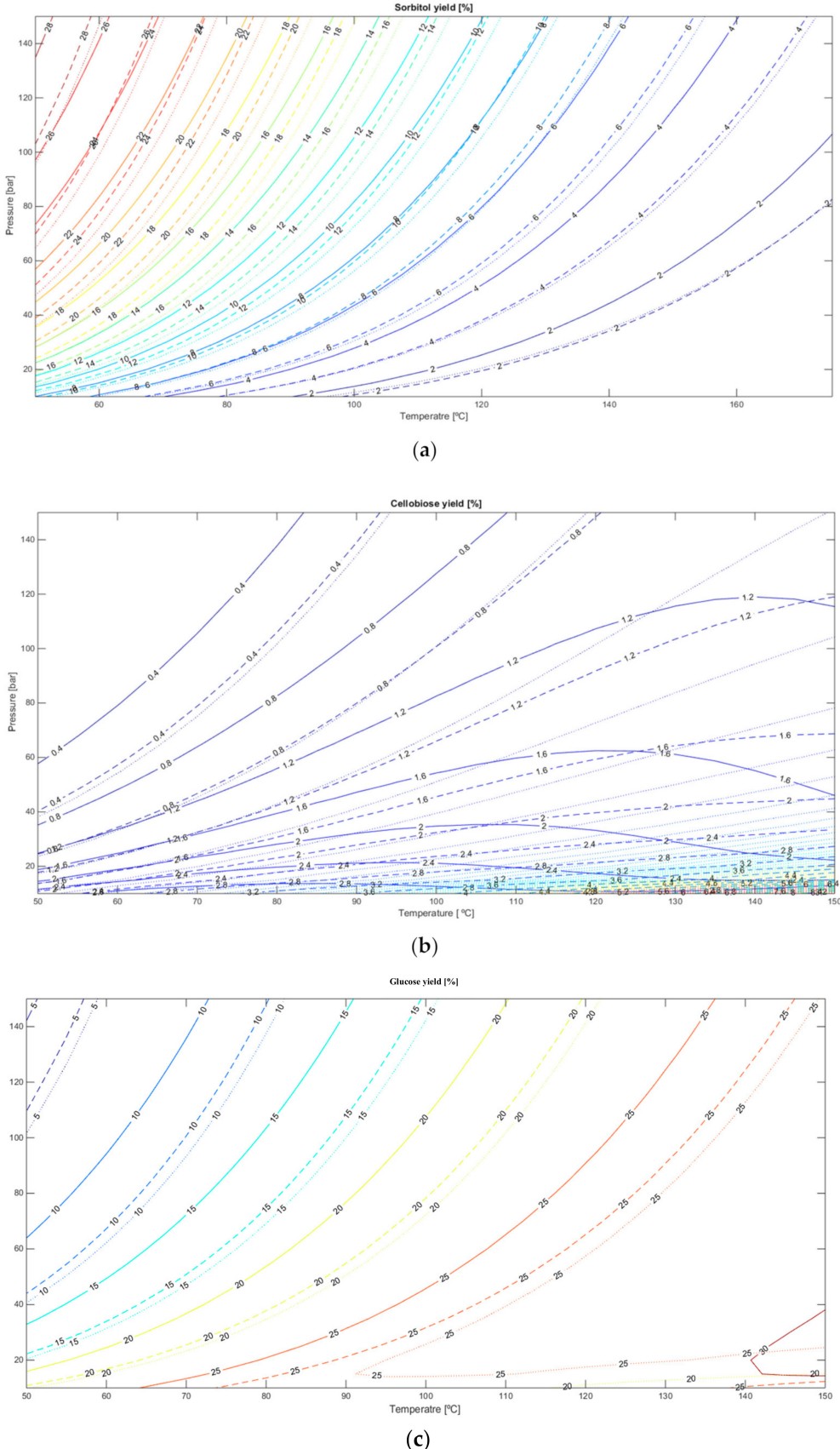

**Figure 6.** Influence of temperature, pressure and H$_2$/WBO molar ratio on (**a**) sorbitol yield, (**b**) cellobiose yield, and (**c**) glucose yield.

However, since bio-oil has a pH < 5, bio-oil-water solutions can be directly hydrolysed/hydrogenated at lower temperatures, as shown by Sanna et al. [46].

In summary, the thermodynamic simulation suggests that temperatures of about 60–90 °C and pressures in the range 40–70 bar would maximise the production of sorbitol from sugars.

Insoluble bio-oil fraction: For the HDO of IBO, there was not a clear reaction pathway, so that the thermodynamic equilibrium was calculated using a Gibbs reactor, which was able to calculate the composition of the reactor output minimising the $\Delta G$. In order to study the temperature effect in terms of product yield, the reaction temperature was varied between 50 and 500 °C at 10 bar selecting a $H_2$/IBO molar ratio = 1. Figure 7 shows the yields of the IBO HDO products. The residual fuel yield decreases according to the increase of temperature, favouring diesel, kerosene and heavy gasoline yield up to about 300 °C, where the diesel yield increases to the expenses of the other products. At 300 °C, it also corresponds the lowest yield of fuel gas. Interestingly, at about 150 °C, the residual fuel can be completely converted, in the same time the maximum yield in term of kerosene and light gasoline are achieved.

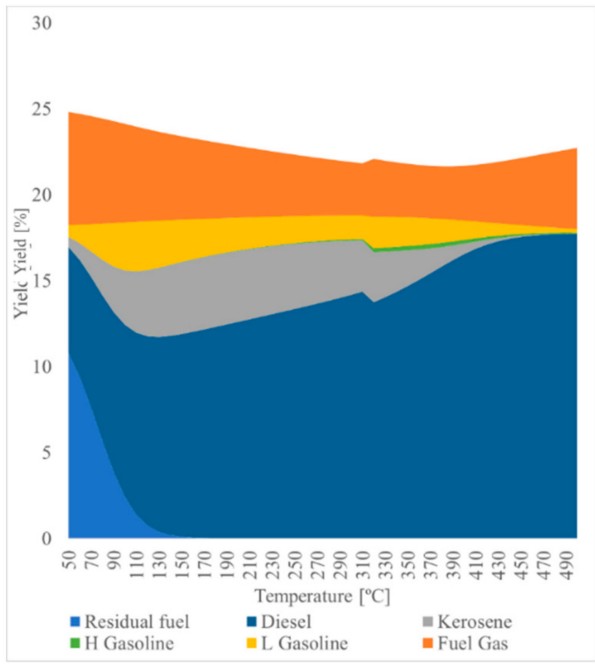

**Figure 7.** Influence of temperature (at 10 bar and feed molar ration $H_2$/IBO = 1) on the HDO products yield.

The effect of pressure (10–150 bar) to the HDO of the IBO was instead studied at 100 °C and using a $H_2$/IBO molar ratio = 1. In that specific set of operating condition, pressure has little influence in terms of products yield, as can be seen in Figure 8. As main products, at 10 bar, the process yields about 13% diesel and about 6% fuel gas. Instead, the influence of the feed/$H_2$ molar ratio at 20 bar and 240 °C (Figure 9) was more evident, in particular for the diesel and kerosene yield. An increment of $H_2$/IBO molar ratio promoted the HDO reactions towards more kerosene at expenses of diesel. This is related to the higher presence of $H_2$, which improves the breaking of C–C bond of compounds with higher molecular weight, aromatic rings and double C–C bonds [29].

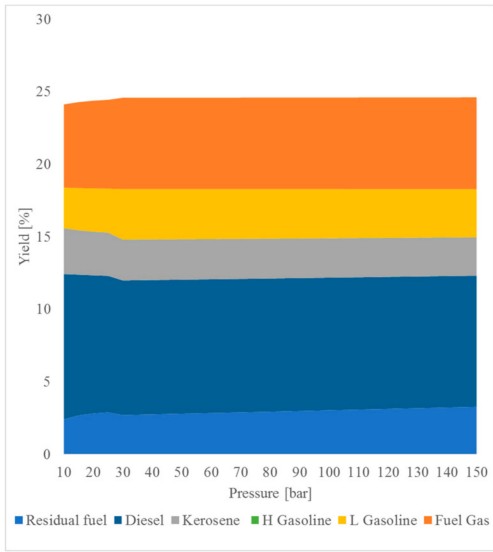

**Figure 8.** Influence of reaction pressure, 100 °C and feed molar ratio H$_2$/IBO = 1 on the HDO products yield.

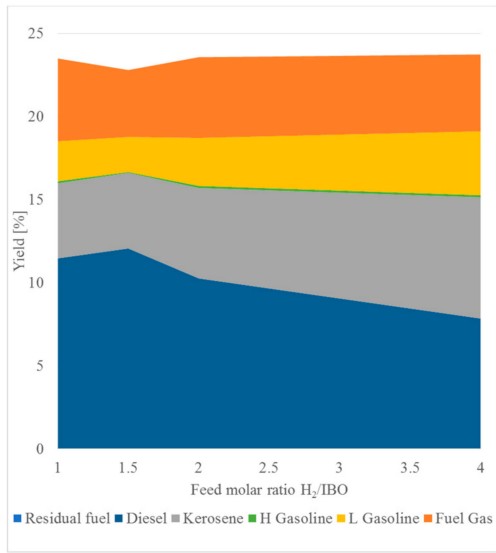

**Figure 9.** Influence of feed molar ratio H$_2$/IBO at 20 bar and 240 °C on the HDO products yield.

In fact, at a feed molar ratio of 4, the diesel yield is at its minimum (8%), while the kerosene (8%) and light gasoline (3.5%) yields are at their maximum.

### 2.3.1. Economic Potential of Second Level

The success of a chemical process depends from its $EP_2$, since the objective is to have products with high added economic value than the raw material. Usually, the raw materials purchase represents from 33% to 85% of total processing costs [20]. The $EP_2$ was dived in two parts, each of which addresses the economic feasibility for the WBO–HDO and IBO-HDO processes, respectively.

Figures 10 and 11 evaluate the $EP_2$ (MM\$/y) varying the pressure, temperature, H$_2$/WBO molar ratio (MR), liquid and vapour (R$_1$, R$_2$) recycle. As shown in the first line of the Figure 10, the R$_1$ and R$_2$ were maintained constant and equal to 0, while the other parameters varied. The $EP_2$ did not change drastically with the reaction pressure, whereas there was a decrement of EP$_2$ when the temperature increased; trend explained by the exothermicity of the reactions involved.

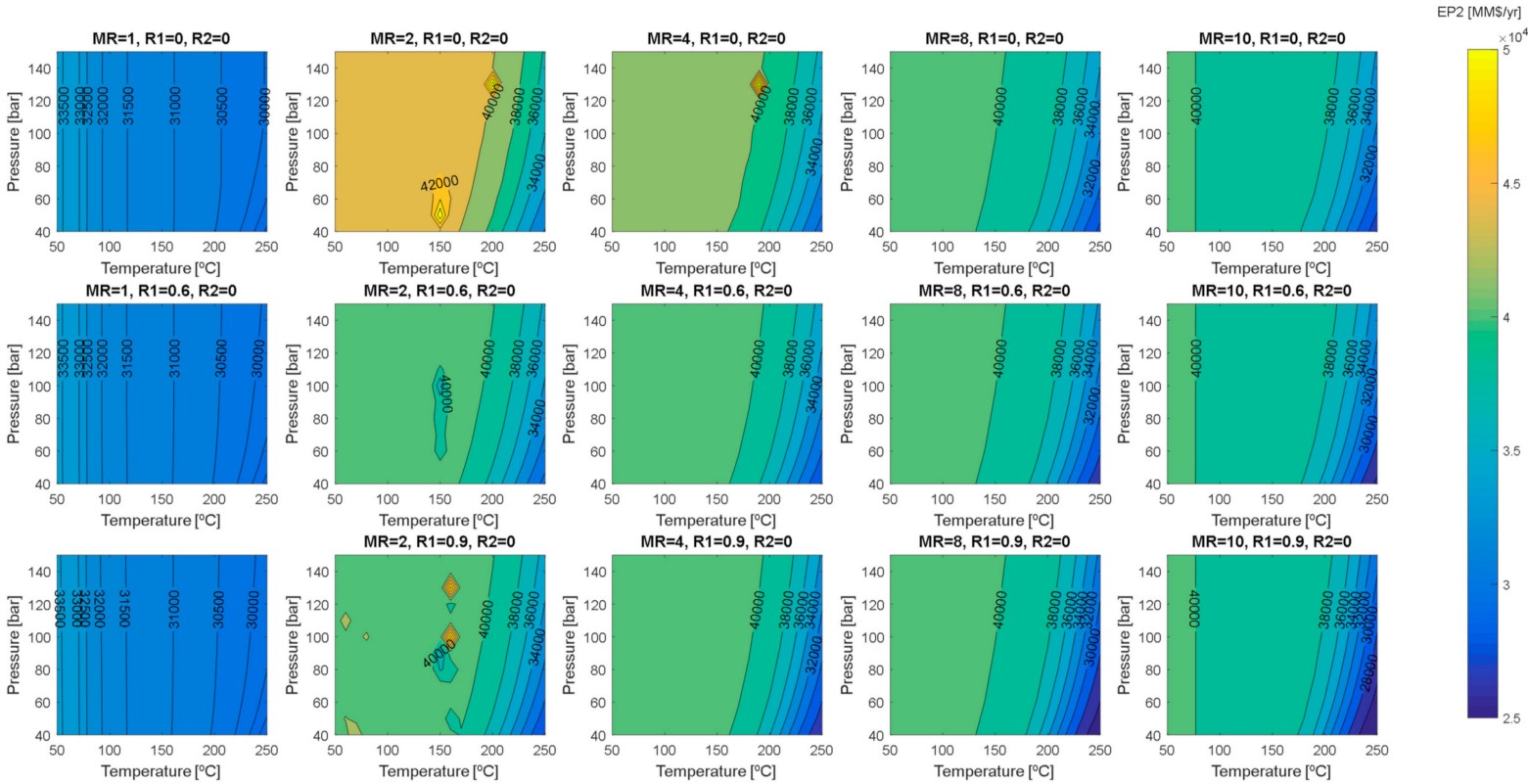

**Figure 10.** Economic potential of the second level [MM$/y] HDO–WBO, varying the operating conditions. Part 1.

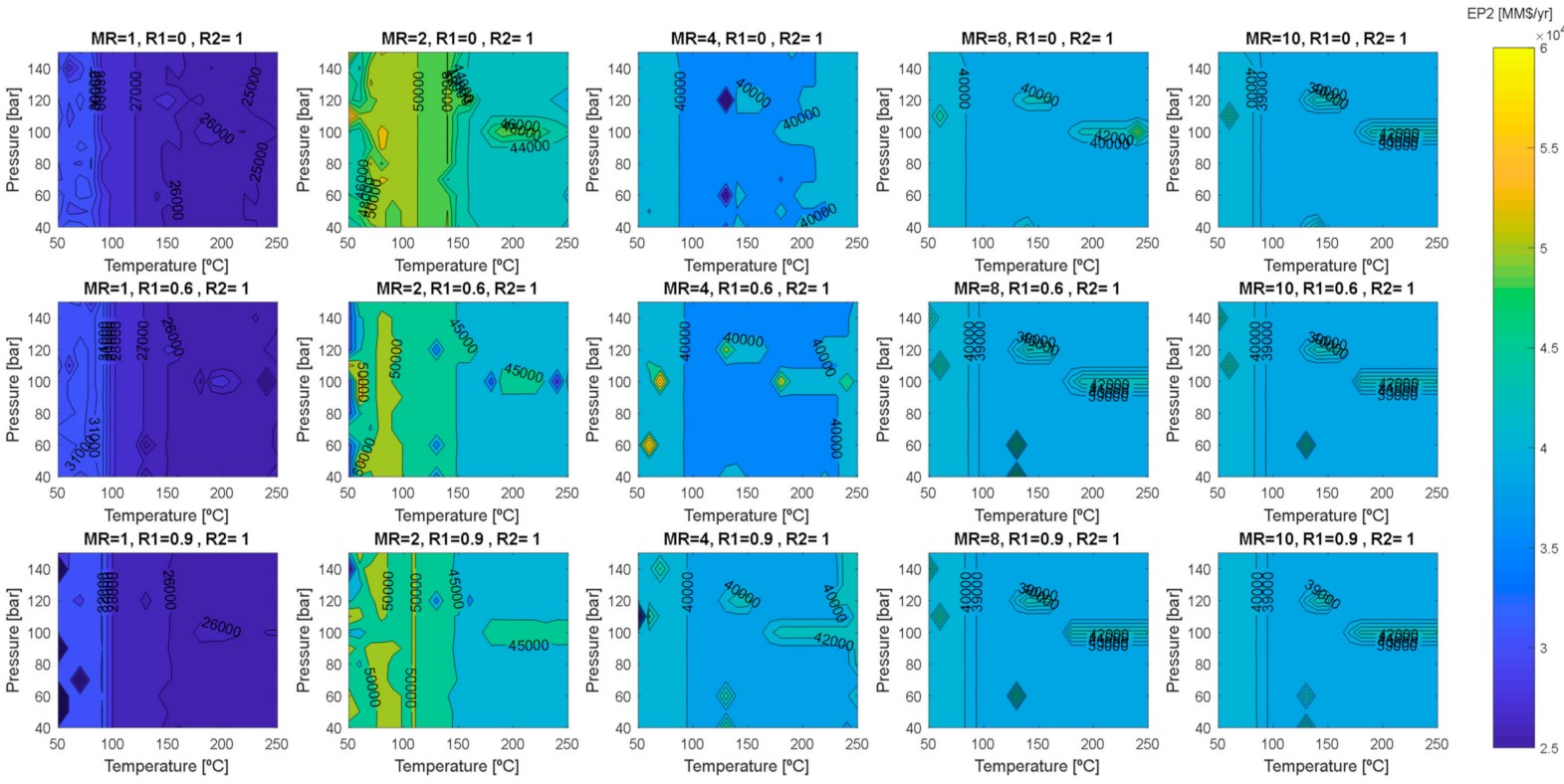

**Figure 11.** Second level economic potential [MM$/y] for the HDO of WBO, varying the operating conditions. Part 2.

Varying the WBO/$H_2$ MR from 1 to 10 the $EP_2$ value varied from ~32,000 MM\$/y to ~38,000 MM\$/y, indicating that very large $H_2$ presence promotes full conversion (or equilibrium), incrementing the $EP_2$ value. The highest value ($EP_2$ ~48,000 MM\$/y) was obtained for a MR = 2. The decrement of $EP_2$ from 48,000 to 38,000 MM\$/y for a MR 2 and 10, respectively was due mainly to the increment of the $H_2$ cost.

Maintaining the same operating condition and varying $R_2$ from 0 to 1, as reported in Figures 10 and 11, the $EP_2$ improved. The increment of $EP_2$ for the highest liquid recycle stream caused the shift of the equilibrium reaction toward the products.

For the IBO–HDO, the operating condition, for having a positive $EP$ resulted to be the following: temperature from about 150 to 275 °C, reaction pressure between 40–150 bar and $H_2$/IBO MR = 1–1.5 with $EP_2$ between 5.5–6 M\$/y (Figure 12). Furthermore, Figure 12 clearly suggests that a temperature lower than 150 °C is undesired in terms of $EP_2$.

$EP_2$ for WBO–HDO resulted in 5.5 ± 0.5 MM\$/y at pressure between 55–150 bar, temperature from 50 to 150 °C and $H_2$/WBO molar ratio between 1 and 2. The above mentioned $EP$ can be only obtained if the reactions products (diols, mono-alcohols etc.) are separated from the water solution and not considered as drop-in fuels, but sold as chemical commodities. Additionally, the $EP$ does not take into consideration separation costs, which can be considerable. Remembering that the data were calculated by thermodynamic analysis, the catalyst and the utilities cost were not considered at this point and the products' distribution was the maximum possible, this preliminary $EP_2$ represented the maximum profit per year of the process.

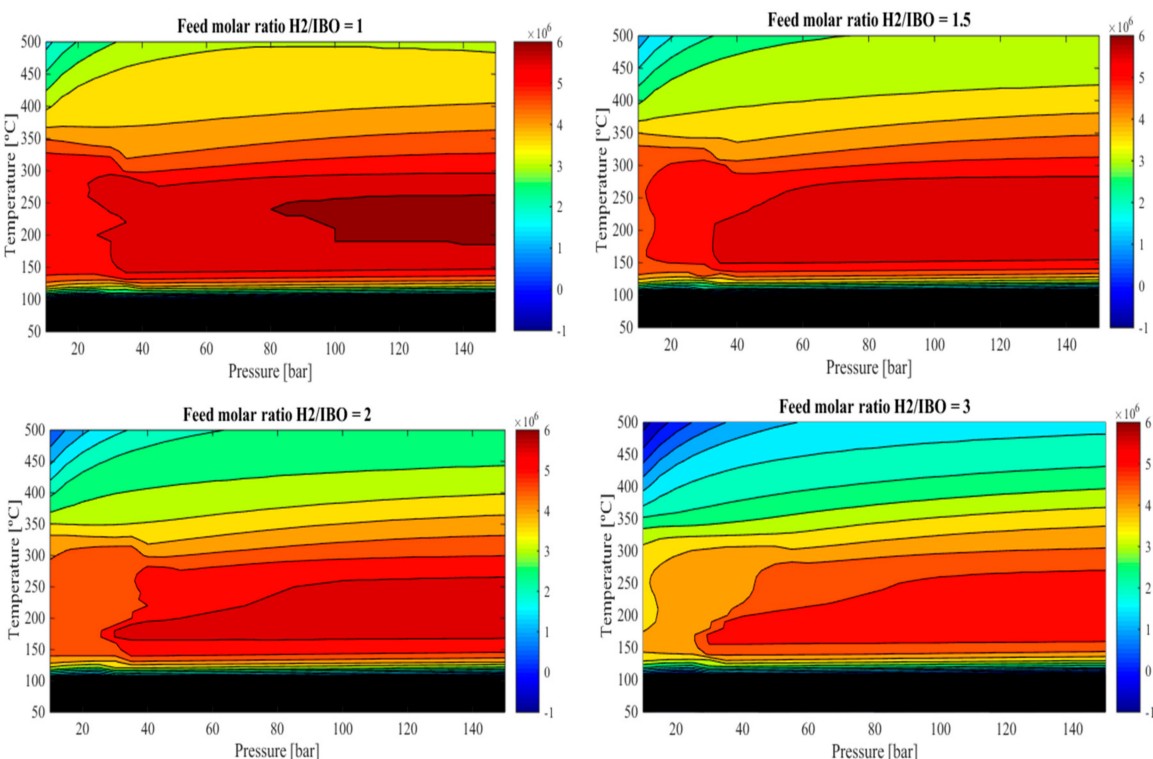

**Figure 12.** Second level economic potential [\$/y] for the HDO of IBO, varying the operating condition.

### 2.4. Level 3: Recycle Structure

Having solved the input-output structure, the simulation passed to a further level of detail in which the necessary recycle streams, the cost of a hypothetical compressor, the reactors number, the possibility to operate them adiabatically and the reactors cost were evaluated in terms of economic potential of third level.

### 2.4.1. Compressor Effect

The compressor cost was calculated using the Guthrie's correlation function of power supplied, while the electricity cost was assumed constant. The $H_2$ was considered supplied at 40 bar from a steam reforming of methane plant [47], Figure 13 shows the relation among the cost to compress the $H_2$ feed varying the reaction pressure (50–150 bar), and the $H_2$/WBO MR (1–10). It is worth to note that there was no reaction pressure limitation for a $H_2$/WBO MR up to 8.

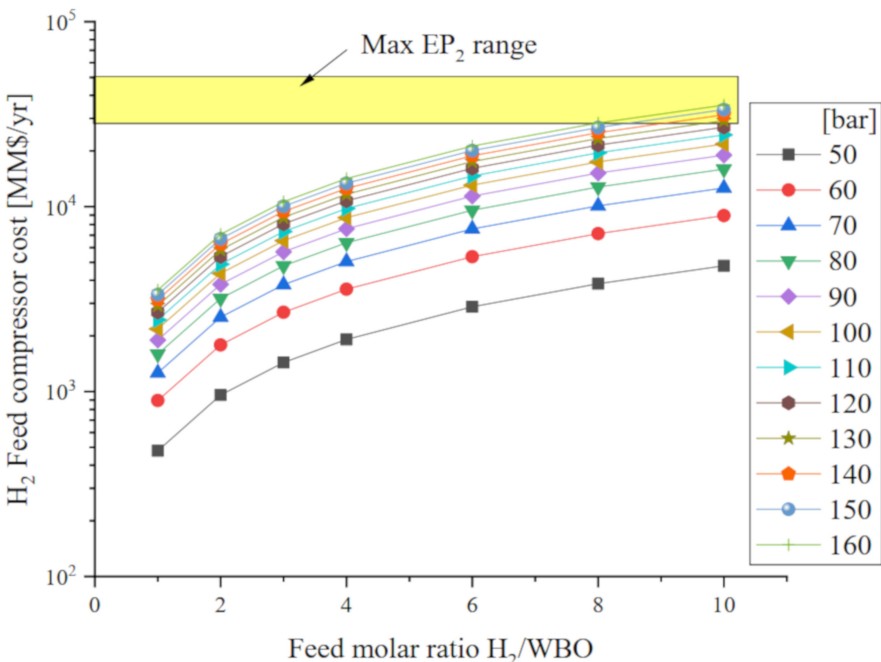

**Figure 13.** Compressor feed cost, varying the reaction pressure and $H_2$-WBO feed molar ratio.

The vapour recycle ($R_1$) was evaluated in order to improve the products yield without penalising the $EP_3$. To maintain the same feed pressure into the recycle stream, a compressor system is expected, due to (i) pressure drop inside the system; (ii) or/and a deliberate pressure decrement in order to separate the liquid from the gas phase.

In our case, the compressor into the recycle stream is due at the presence of the flash unit to separate the mixture into two phases. Figure 14 shows the cost related to recycle the vapour and liquid phase varying the pressure difference ($\Delta P$) between the $R_1$ and the liquid recycle stream ($R_2$). Maintaining $R_2$ constant, for $R_1 > 0.8$ and $\Delta P > 10$ bar, the compressor cost are equal to the $EP_2$, not admissible for the realisation of the process.

While, varying only $R_2$, there is a decrement of compressor cost, due a decrement at a low amount of vapour recycle stream.

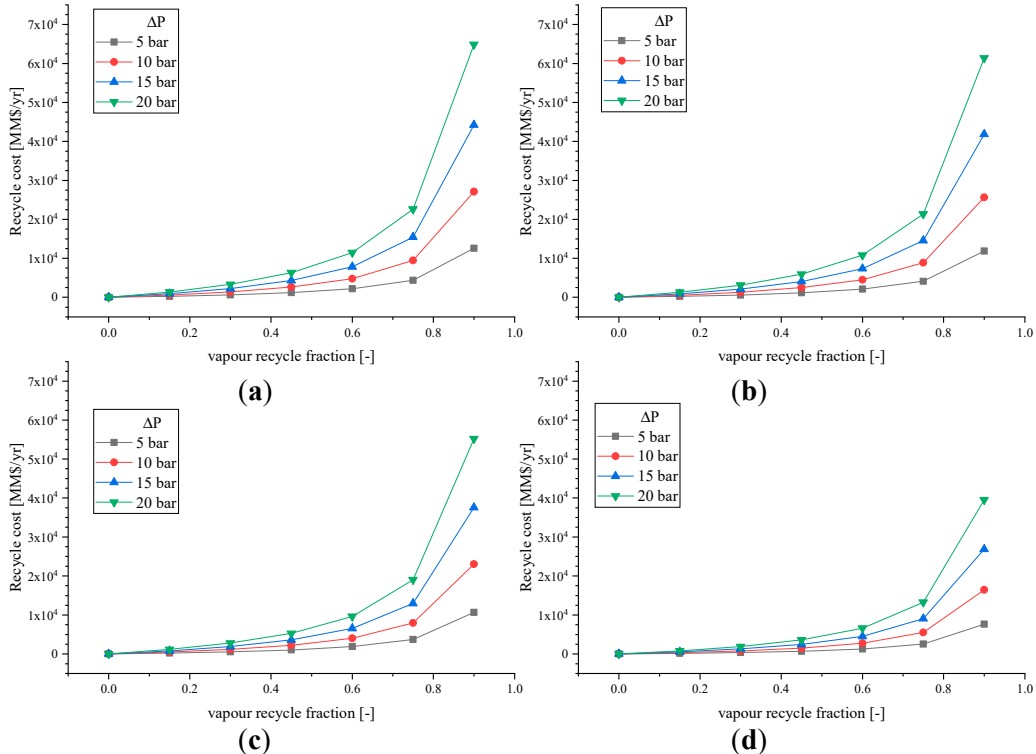

**Figure 14.** Recycle cost for HDO of WBO for MR = 2 in function of vapour recycle fraction, varying $\Delta P$ and the liquid recycle fraction ($R_2$) at (**a**) $R_2$ = 0, (**b**) $R_2$ = 0.3 (**c**) $R_2$ = 0.6, (**d**) $R_2$ = 0.9.

### 2.4.2. Reactor Heat Effect

This analysis was carried out to decide whether the reactor had to be operated adiabatically, with direct heating or cooling, or whether a diluent or heat carrier was needed.

The process is average exothermic for HDO–WBO and the presence of a recycle could drastically improve the reaction temperature, as reported in Figure 15, and for a reactor inlet temperature of 100 °C, a $H_2$/WBO MR = 2 and 50 bar. The cited figure shows how the temperature into the reactor output stream changed varying the sorbitol yield, liquid and vapour recycle. For all the cases, an increment of R1 improved the reaction temperature and decremented the sorbitol presence due, additionally improving the diols production. While an increment of $R_2$ decreased the temperature into the reactor and also increased the sorbitol yield. Then, $R_2$ had the principally function of a thermal carrier. To remedy the high temperature, adding water as a thermal carrier into the reactor was taken into consideration.

### 2.4.3. Economic Potential of 3rd Level

To obtain the $EP_3$, $EP_2$ data were updated using Equations (14) and (15), by including the cost related to the reactor, compressor and heat exchange units, with the relative utilities to achieve the operating condition.

For the WBO–HDO, the reactor system consisted into two adiabatic packed bed reactors in the presence of 5 wt % Ru/C as catalyst (void bed fraction equal to 0.40) with intermediate cooling. The inlet stream was preheated up to 175 °C using water as thermal carrier, imposing a max temperature in the outlet of 200 °C. Figure 16 shows the $EP_3$ for the WBO-HDO varying $H_2$/WBO MR, reaction pressure and also gas ($R_1$) and liquid ($R_2$) recycle fractions.

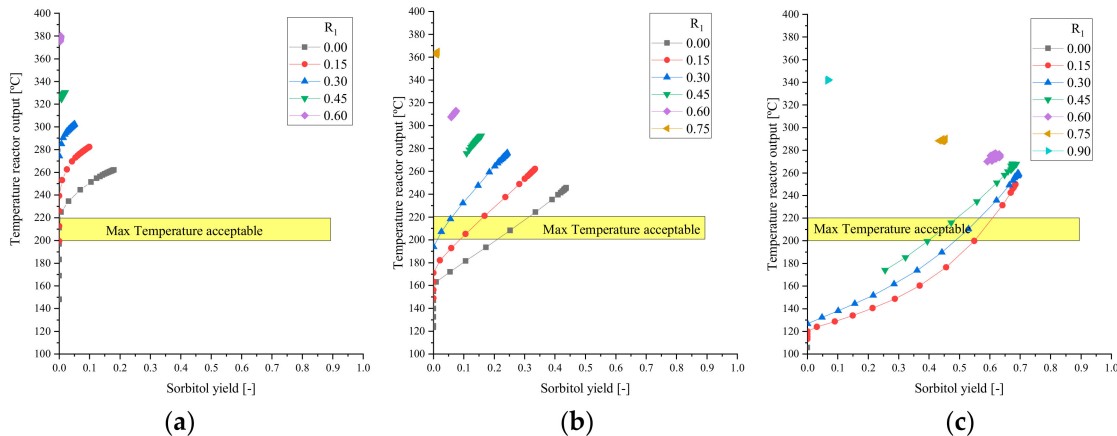

**Figure 15.** Reactor exit temperature vs. sorbitol yield, varying the vapour recycle fraction ($R_1$), at (**a**) $R_2 = 0$, (**b**) $R_2 = 0.6$ and (**c**) $R_2 = 0.9$.

Analysing the effect of $R_2$ into $EP_3$, there was an evident decrement of the profit, mainly due to an increment of the reactor size and also the utility relate to preheating $R_2$.

An increment of the $EP_3$ was noted by increasing the $H_2$/WBO MR up to 2, where the $EP_3$ achieved the maximum value. For MR > 2, the cost related to the compressor unit downstream of the reactor increase resulted in a decrease of the $EP_3$. Furthermore, an increment of the reaction pressure enhanced the $EP_3$ in all the cases studied except for $R_2 = 0.9$; while a drastic decrement of $EP_3$ was noted for $R_1 > 0.4$.

The third level of WBO–HDO fraction resulted in an $EP_3$ of 40,600 M\$/y, for a temperature of 250 °C at 100 bar, $R_1 = 0.4$ and in absence of $R_2$.

The stages number in the IBO–HDO process was imposed equal to 3, with the target to minimise the catalyst (Co-Mo/$\gamma$Al$_2$O$_3$) amount loaded. The process conditions were 250 °C and 50 bar. The output stream was cooled down to 35 °C. The separation systems were considered ideal and evaluated at fourth level. For the IBO-HDO section the cost were calculated using the guideline from Gary et al. [48], achieving a 5.99 M\$/y.

The $EP_3$ for the IBO-HDO section resulted negative (−0.19 M\$/y), suggesting that this section of the process is not competitive with the actual fuel price based on market price. With the target to achieve the break-even point for IBO–HDO, the minimum fuel selling price (MFSP) was instead assessed:

$$MFSP_i = \left.\frac{Annual\ plant\ cost}{EP_2}\right|_{IBO} \cdot Fuel\ price_i. \tag{8}$$

Then, to be economically competitive, the fuel market price should be higher than the MFSP.

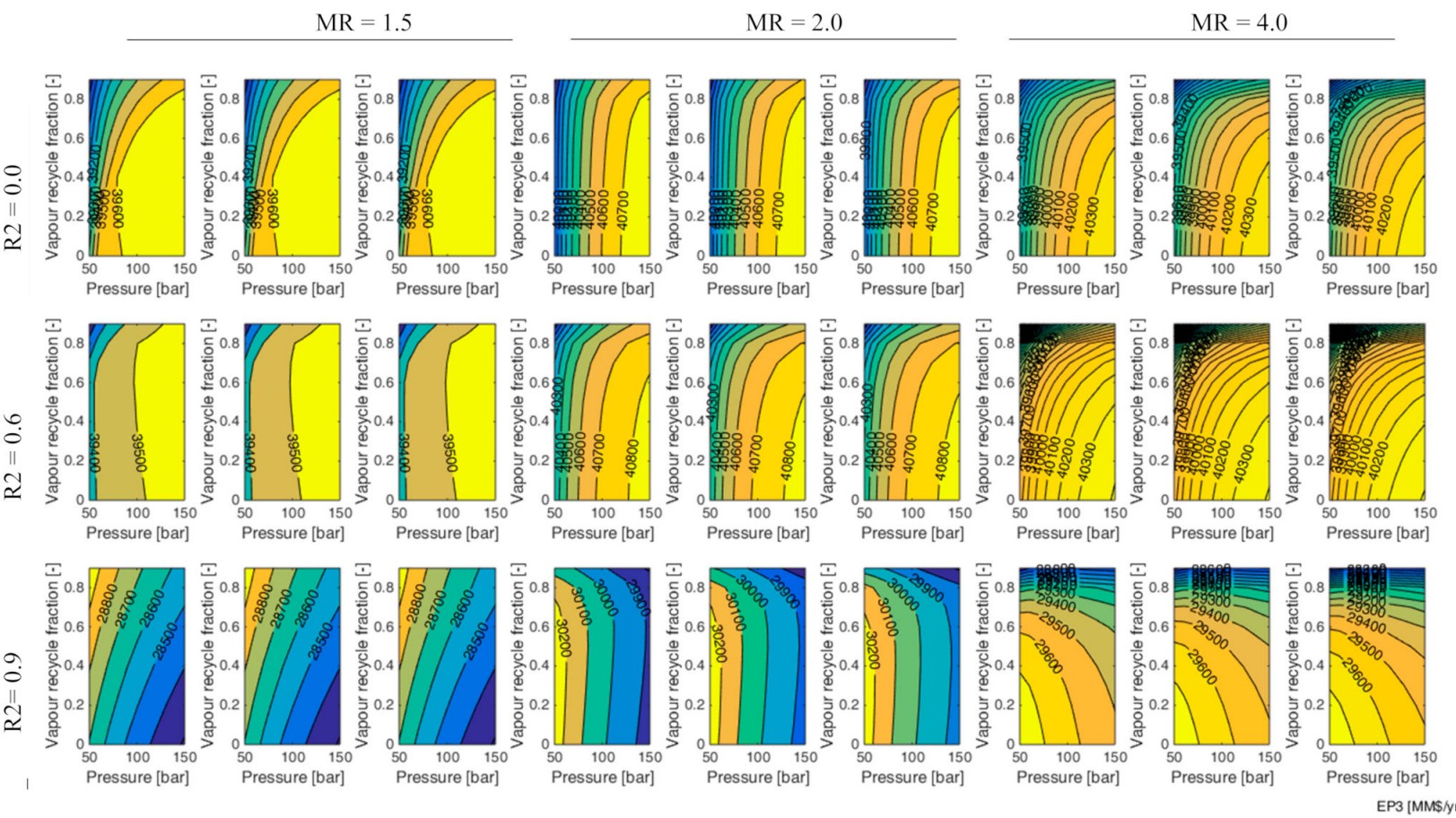

**Figure 16.** Economic potential of the third level WBO varying the operating condition.

### 2.5. Level 4: Separation System

In level 4, the separation system required to recover products with high purity was designed. For the WBO, the outlet stream of the reactor system (see Figure 17) was initially separated by a flash unit (S-401) able to split the two phases (vapour and liquid phases) and allow their purification by different methods.

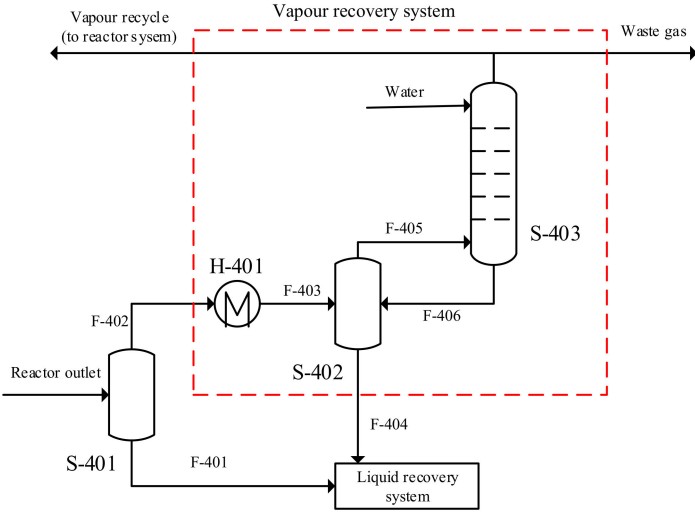

**Figure 17.** Recovery vapour system.

The vapour stream (F-402) was cooled down to 35 °C using cooling water as utility stream. Since a simple cooling is not able to guarantee an efficient recovery of the products in the vapour phase, solvent absorption was considered. The solvent was based on cost, affinity and volatility difference with the solute. Simplifying hypotheses were made: (i) the thermal effects were neglected, operating at a temperature equal to that of the gas to be treated and at approximately atmospheric pressure; (ii) hydrogen and methane were assumed to remain in the gas phase; (iii) the absorbing recovery of methanol was set at 99.5%, (iv) the absorber system was designed according to Kremser [49].

Therefore, to recover the condensable products from F-405, a water-based absorber unit (S-403) was used.

The use of an absorber minimises the products lost into the vapour phase. The choice of a vapour recovery unit derives from a compromise between the recovery efficiency and the economy of the process. As reported in Table 2, the presence of the vapour recovery system results in a cost decrement of than 50% for all the scenarios.

**Table 2.** Absorber system cost at various inlet and outlet pressures.

| | | Absorber System Cost + Product Lost [k$/y] | | Products Lost w/o Absorber System [k$/y] | |
|---|---|---|---|---|---|
| | | Pressure Outlet [Bar] | | Pressure Outlet [Bar] | |
| | | **10** | **20** | **10** | **20** |
| Pressure inlet [bar] | 100 | 1050 | 564 | 2820 | 1770 |
| | 110 | 950 | 547 | 2740 | 1650 |
| | 120 | 880 | 474 | 3180 | 1340 |
| | 130 | 1040 | 531 | 2850 | 1630 |
| | 140 | 100 | 522 | 2590 | 1600 |
| | 150 | 1010 | 606 | 2560 | 1830 |

The liquid phase from the flash S-401 and the liquid stream from the vapour recovery system (F-402) were sent to a liquid separation system. Heuristic rules were followed for the design: (i) the first

separation targeted the components in greater quantity and with higher volatility; (ii) the most difficult separations were carried out at the end.

A total of 77.46 ton/day was fed the liquid recovery system, which consisted in a water and complex mixture of organic compounds (see Table 3). For the first separation, the distillation system was rejected due to the glucose TBP being higher than its realistic polymerisation temperature.

**Table 3.** Composition (wt %) input stream to liquid recovery system.

|  | **Mass Fractions** | **TPB [°C]** |
|---|---|---|
| Ethanol | 0.027 | 78 |
| Water | 0.336 | 100 |
| Hydroxyacetaldeyde | 0.003 | 145 |
| Ethyl cyclohexanol | 0.029 | 191 |
| Ethylene glycol | 0.046 | 197 |
| Cyclohexane ethanol | 0.001 | 210 |
| Propandiol | 0.127 | 214 |
| Benzendiol | 0.042 | 222 |
| Butanediol | 0.015 | 227 |
| Hydroxypropionic acid | 0.015 | 285 |
| Vanillin alcohol | 0.029 | 290 |
| Glucose | 0.018 | 344 |
| Sorbitol | 0.310 | 431 |

Therefore, a simulated moved bed (SMB), a typical application in the sugar processing industry, was considered to separate glucose from the stream F-407 [50]. Eight packet columns filled with silica were therefore modelled (See Figure 18). In the system, it is possible to define a first zone (from SMB-1 to SMB-3), where the sorbitol was recovered from the solid phase into the extract stream. In the second zone (SMB-4 and SMB-5), the solid phase absorbed the sorbitol coming from the feed, while in the third zone (from SMB-6 to SMB-7), the stream was enriched in glucose. All the four fluid streams were periodically switched forward one column position, causing stepwise movement of the zones. The SMB system was able to separate the totality of the glucose and recover 98% sorbitol (45% purity) into the extract stream. The cost of the SMB system was estimate equal to 34 $/kg$_{feed}$ [51].

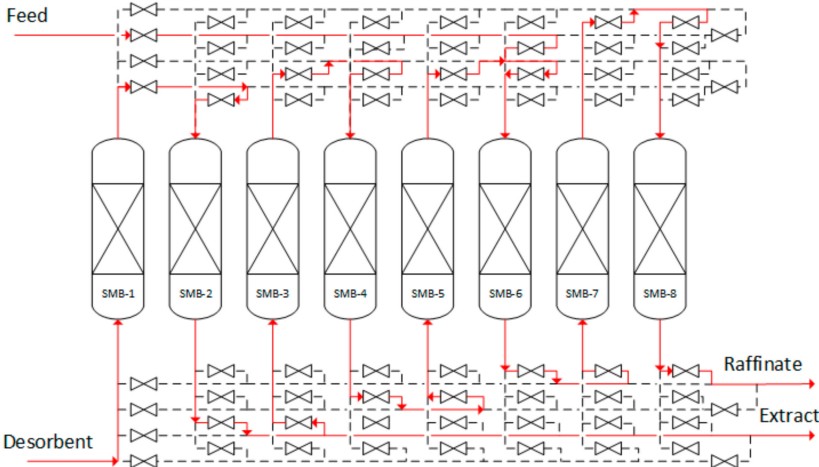

**Figure 18.** Simulated moving bed for liquid recovery system.

The stream rich in sorbitol (without glucose) was then purified by a distillation system (see Figure 19, Table 4), which resulted in an overall cost of 704.8 M$/y.

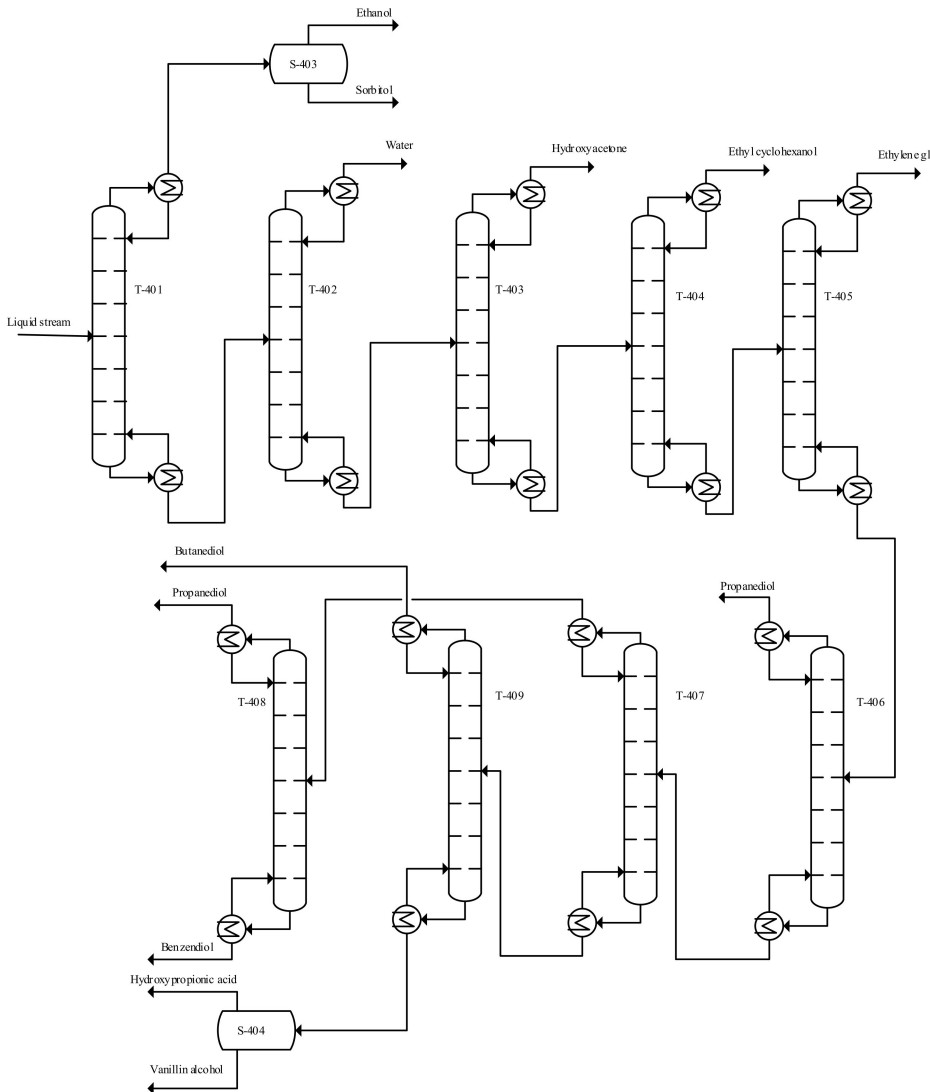

**Figure 19.** Distillation system for the separation of WBO products from HDO.

**Table 4.** Distillation columns specification.

|  | T-401 | T-402 | T-403 | T-404 | T-405 | T-406 | T-407 | T-408 | T-409 |
|---|---|---|---|---|---|---|---|---|---|
| Reflux ratio [-] | 0.525 | 0.405 | 0.434 | 92.1 | 6.95 | 5.59 | 11.5 | 11.6 | 0.60 |
| N° stages [-] | 100 | 20 | 92 | 230 | 146 | 224 | 816 | 442 | 52 |
| Feed stage [-] | 58 | 22 | 66 | 146 | 94 | 142 | 624 | 132 | 34 |
| Reboiler heating required [Gcal/h] | 334.9 | 92.8 | 2.1 | 387.8 | 146.0 | 239.4 | 208.9 | 82.5 | 9.0 |
| Condenser cooling required [Gcal/h] | 246.2 | 18.87 | 0.02 | 124.8 | 126.7 | 202.5 | 191.3 | 91.35 | 7.70 |
| Distillate temperature [°C] | 117.0 | 100.0 | 138.3 | 166.0 | 196.6 | 213.6 | 218.6 | 213.6 | 227.4 |
| Bottom temperature [°C] | 125.3 | 208.2 | 211.7 | 213.3 | 220.2 | 230.4 | 257.5 | 222.3 | 312.8 |
| Capital cost [M$/y] | 3.92 | 0.70 | 0.68 | 14.70 | 7.02 | 14.32 | 52.17 | 28.26 | 0.42 |
| Utility cost [M$/y] | 131.51 | 32.44 | 0.70 | 139.30 | 58.90 | 96.16 | 85.09 | 34.88 | 3.62 |

While, the distillation was the separation technique used to achieve the commercial standard of purification of HDO–IBO products. The separation sequence was designed, removing first of all the compounds with lower BP. The distillation tower was imposed a plate efficiency equal at

50% and for mechanical stability the height/diameter ratio considerate was between 20 and 30, and the greater height was 50 m. The overall cost of this plant section was equal 2.94 M$/y, based on Gary et al. [48] correlation.

## 2.6. Economic Evaluation

The cost of the separation system was the final stage for allowing the calculation of the final *EP* (or $EP_4$). The $EP_4$ for the HDO–WBO was obtained subtracting from the $EP_3$ (40,070 M$/y) the cost related to the vapour recovery system (1 M$/y), the SMB unit (704.8 M$/y) and the distillation system (1130 M$/y). The $EP_4$ of the WBO resulted positive and equal to 38,234 M$/y, indicating that the HDO–WBO process would be competitive for the production of chemicals. Furthermore, an advantage of converting the WBO to chemicals is related to the large number of products, with the possibility to orientate the production through the compounds demand. The actual plant produces 24.0 ton/day of sorbitol, but an increment of its yield could be further achieved converting the 1.39 ton/day of unconverted glucose. Additionally, this work was able to produce 9.84 ton/day of propanediol at the actual market price, 1.27 $/kg [52], which resultingly is more economically feasible than other bioprocesses such as the microbiological conversion of glycerol, with an estimated price of 2.43 $/kg [53].

Instead, the $EP_4$ calculated for the HDO of the IBO resulted negative and equal to 3.13 M$/y, which is not economically competitive based on market fuels price.

Equation (8) calculated the MFSP (and reported in Table 5) in order to have a profitable HDO–IBO process. The MFSP varying from 0.406 to 1.465 $/kg for the different fuels produced, 53.9% higher than the fossil fuel. The MFSP obtained did not differ from the data of Carrasco et al. [18] estimating a MFSP equal to 1.38 $/L from hydrotreatment process of pyrolytic oils, for a capacity of 2000 ton/day of residual forest biomass. With a total cost of investment (TCI) equal to 171.5 MM$ Furthermore, Wright et al. [16] hydro-processed 1440 ton/day bio-oil, able to produce 5182 barrels per day of naphtha and diesel range blend fuel the MFSP was 0.546 $/L, which was in the same range of MFSP obtained in this work.

**Table 5.** Minimum fuel sell price for HDO of IBO.

| Compound | Price from Fossil Fuel [$/GJ] | Mass Flow [Ton/Day] | MFSP | |
|---|---|---|---|---|
| | | | [$/GJ] | [$/kg] |
| Fuel gas | 3.72 | $1.73 \cdot 10^{-4}$ | 5.71 | 0.685 |
| Light gasoline | 14.44 | 125.38 | 22.22 | 0.966 |
| Heavy gasoline | 21.88 | 42.3 | 33.69 | 1.465 |
| Kerosene | 11.92 | 20.8 | 18.35 | 0.791 |
| Diesel | 12.26 | 130.7 | 18.87 | 0.816 |
| Residual fuel | 6.82 | 9.08 | 10.50 | 0.406 |

The *EP* result is able to represent the profit of the whole process, taking in consideration the cost of all the operation units, the utility streams, and feeds. Furthermore, the *EP* has been used to calculate the costs associated to the plant (piping, instrumental control, maintenance and repairs, etc.), as reported in Table 8.

The HDO process of pinewood bio-oil was designed to process 10 Mton/y with a TCI of 55,271 MM$ and profit after the tax equal to 9406 MM$ per year. Since the parameters cited above do not give a real potential of the plant, the ROI and the pay-out of the investment were evaluated. In particular, a ROI of 69.18% and a recovery of the initial investment after 2.48 years was obtained.

To further evaluate the economic feasibility of the plant, the DCFROR for a plant life of 20, 25 and 30 years was calculated, resulting in a rating of 18.75%, 19.02% and 19.11%, respectively. This is 3.95 times the interest if TCI was deposit in the bank (4.75%) [54].

## 3. Methods

Therefore, a suitable simulation model for an industrial bio-oil hydrogenation plant was developed in order to predict the behavior of the reactions during the upgrading process. In this regard, first, a reaction network responsible for the hydrogenation of the bio-oil is proposed and then, physical and empirical correlations are applied.

To design the hydrogenation of bio-oil the heuristic method described by Douglas et al. [20] was used. This method provides solving the problems by different detailed layers, starting from a basic level, to a level where more specific knowledge is required. Each level has been evaluated using the Economic Potential (*EP*) that indicates the annual profit of the process, depending on the variable of projects and the specific required of the level:

Level 0: Preliminary information: The target of this level is to find all the information present in literature about the process, including the reactions involved, the catalysts studied and operating condition at which the products can degrade. In order to evaluate the system, the following variables have been defined:

$$Yield_i = \frac{mass\ flow\ of\ i-component}{mass\ flow\ of\ Bio-oil}\ [-], \tag{9}$$

$$MR = \frac{molar\ flow\ of\ H_2\ feed}{molar\ flow\ of\ Biooil\ fraction}\ [-], \tag{10}$$

$$R_1 = \frac{malar\ flow\ rate\ of\ vapour\ recyvcled}{malar\ flow\ rate\ of\ vapour\ reactor\ output}\ [-], \tag{11}$$

$$R_2 = \frac{malar\ flow\ rate\ of\ liquid\ recyvcled}{malar\ flow\ rate\ of\ liquid\ reactor\ output}\ [-]. \tag{12}$$

Level 1: Batch vs. continuous process. At this level, the process is defined as how to operate, continuous or batch.

Level 2: Input-output structure. At this level, the process is considered as a black box, with input and output streams. By the material balance, the products have been calculated using the thermodynamic data. Moreover, level 2 corresponds to the maximum *EP* obtainable calculated as:

$$EP_2 = Product\ value + Byproduct\ values - Raw\ material\ cost\ [=]\$/yr. \tag{13}$$

The cost of each compound is reported in Table 6.

Level 3: Recycle structure. By using the kinetic rates, it was decided how the reactor has to work and the possibilities in terms of recycle streams. The $EP_3$ is composed of $EP_2$ plus the addition of the reactions cost:

$$EP_3 = EP_2 - L_3\ cost\ [=]\$/yr, \tag{14}$$

$$L_3\ cost = reactor\ cost|_{L_3} + compressor\ cost\Big|_{L_3} + heat\ excange\ cost\Big|_{L_3} + utitlities\ cost|_{L_3}\ [=]\$/yr. \tag{15}$$

The reactor cost was calculated by Guthrie's correlation, where all the equipment costs were actualised in 2016 by the Marshall and Shift index (M&S).

Level 4: Separation system. The most suitable equipment to obtain the desired products and to recover them with a purity that is as high as possible were found. To determine the general structure of the separation system, the phase in output of the reactor was specified. The separation system was split into two sections, vapour and liquid phase, which affected the $EP_4$:

$$EP_4 = EP_3 - L_4\ cost\ [=]\$/yr, \tag{16}$$

$$L_4\ cost = liquid\ sep.\ cost\Big|_{L_4} + vapour\ sep.cost\Big|_{L_4} + product\ lost\Big|_{L_4} + utitlities\ cost|_{L_4}\ [=]\$/yr. \tag{17}$$

**Table 6.** Chemical price of main reactants and products from bio-oil HDO.

| Compound | Unit | Price | Ref. | Compound | Unit | Price | Ref. |
|---|---|---|---|---|---|---|---|
| Acetic acid [§] | $/kg | 0.839 | [52] | Hydroxyacetone [§] | $/kg | 1.25 | [52] |
| Benzendiol [§] | $/kg | 1.50 | [52] | Hydroxypropionic acid [§] | $/kg | 10.12 | [52] |
| Bio-Oil * | $/GJ | 16.84 | [55] | Kerosene * | $/GJ | 11.92 | [56] |
| Butyric acid [†] | $/kg | 1.21 | [52] | Levulinic acid [§] | $/kg | 5.80 | [52] |
| Butanediol [§] | $/kg | 1.71 | [52] | Light Gasoline * | $/GJ | 14.44 | [56] |
| Diesel * | $/GJ | 12.26 | [56] | Methanol [§] | $/kg | 0.315 | [52] |
| Ethanol [§] | $/kg | 0.67 | [52] | o-Methoxyphenol [§] | $/kg | 1.50 | [52] |
| Ethylene glycol [§] | $/kg | 1.43 | [52] | Propanediol [§] | $/kg | 1.27 | [52] |
| Fuel gas * | $/GJ | 3.72 | [56] | Residual fuel * | $/GJ | 6.82 | [56] |
| Glucose [§] | $/kg | 0.309 | [52] | Sorbitol [§] | $/kg | 1.72 | [52] |
| Heavy Gasoline * | $/GJ | 21.88 | [56] | Methoxy-Hydroxybenzaldehyde [§] | $/kg | 12.0 | [52] |
| Hydrogen * | $/GJ | 12.50 | [57] | Vanillin alcohol [§] | $/kg | 44.46 | [52] |
| Hydroxyl acetaldehyde [§] | $/kg | 1 | [52] | γ-Valero lactone [§] | $/kg | 14.43 | [52] |

* 2015, [†] 2001, [§] 2006.

Cost estimating. Having defined all parts of the process, the economic evaluation was done following the guideline of Douglas et al. [39] and Peters et al. [58], as summarised in Tables 7 and 8. Those values were then used to calculate the return of investment (ROI), which made it possible to calculate the profitability of the overall process:

$$ROI = \frac{Annual\ Profit}{Total\ Investiment}[\%].\qquad(18)$$

The pay-out time, a parameter able to indicate the years for recovering the initial investment:

$$Payout\ time = \frac{Fixed\ Cap. + Start-up}{Profit\ after\ Taxes + Deprec.}.\qquad(19)$$

**Table 7.** Estimation of capital investment cost for the overall system.

| | |
|---|---|
| Direct Costs: | **32,839 MM$** |
| Purchased equipment (22.9% FCI) | 10,758 MM$ |
| Installation, including insulation and painting (8.3% FCI) | 3899 MM$ |
| Instrumentation and controls, installed (6.4% FCI) | 3007 MM$ |
| Piping, installed (7.3% FCI) | 3430 MM$ |
| Electrical, installed (4.6% FCI) | 2161 MM$ |
| Buildings, process and auxiliary (4.6% FCI) | 2161 MM$ |
| Service facilities and yard improvements (13.8% FCI) | 6483 MM$ |
| Land (1% to 2% FCI) | 940 MM$ |
| Indict costs: | **13,765 MM$** |
| Engineering and supervision (9.2% FCI) | 4322 MM$ |
| Construction expense and contractor's fee (12.8% FCI) | 6013 MM$ |
| Contingency (7.3% FCI) | 3430 MM$ |
| Fixed-capital investment (FCI) = direct costs + indirect costs | **46,980 MM$** |
| Working capital (15% TCI) | **8291 MM$** |
| Total capital investment (TCI) = fixed-capital investment + working capital | **55,271 MM$** |

Additionally, the discount of cash flow rate of return (DCFROR), a parameter to estimate the profit as interest index (*i*), gave a direct comparison with bank interest. The DCFROR was calculated, equating the value of the investment value (IPV) to build-up the plant (calculated in 3 years) and the

return value (RPV), calculated in N-years equal to the process service life, assuming different case: 15, 20, 25 and 30 years.

$$IPV = \sum\nolimits_{j=0}^{3}\left\{Work.Cap. + \ Start\_up + \left[a_j{\cdot}TCI{\cdot}(1+i)^j\right]\right\}, \tag{20}$$

$$RPV = \frac{Work.Cap. + Salv.Val.}{(1+i)^N} + \sum\nolimits_{j=1}^{N}\left[\frac{b_j{\cdot}CashFlow}{(1+i)^j}\right], \tag{21}$$

where $a_j$ is the fraction of TCI for different year, $a_1 = 0.1$, $a_2 = 0.4$, $a_3 = 0.4$ and $a_4 = 0.1$; while $b_j$ used to correct the annual cash flow, considering constant after the third year ($b_1 = 0.6$ $b_2 = 0.9$ $b_3 = 0.95$, $j > 4$ $b_j = 1$).

**Table 8.** Estimation of the total product cost for the overall system.

| | |
|---|---|
| Manufacturing cost = direct production costs + fixed charges + plant overhead costs | **55,910 MM$** |
| Direct production costs: | **42,847MM$** |
| Raw materials (35% TPC) | 22,861 MM$ |
| Operating labour (10% TPC) | 6532 MM$ |
| Direct supervisory and clerical labour (10% to 25% of operating labour) | 653 MM$ |
| Utilities (15% of total product cost) | 9797 MM$ |
| Maintenance and repairs (2% to 10% of fixed–capital investment) | 810 MM$ |
| Operating supplies (0.5% to 1% FCI) | 235 MM$ |
| Laboratory charges (10% to 20% of operating labour) | 653 MM$ |
| Patents and royalties (0% to 6% TPC) | 1306 MM$ |
| Fixed charges: | **65,320MM$** |
| Depreciation (10% FCI) | 4698 MM$ |
| Local taxes (3% FCI) | 1409 MM$ |
| Insurance (0.7% FCI) | 329 MM$ |
| Rent (10% of value of rented land and buildings) | 94 MM$ |
| Plant—overhead costs (10% TPC); includes costs for the following: general plant upkeep and overhead, payroll overhead, packaging, medical services, safety and protection, restaurants, recreation, salvage, laboratories, and storage facilities. | **6532MM$** |
| General expenses: | **9406MM$** |
| Administrative costs (15% operating labour) | 980 MM$ |
| Distribution and selling costs (11% TPC) | 7185 MM$ |
| Research and development costs (5% TPC) | 3266 MM$ |
| Financing (5% TCI) | 3266 MM$ |
| Total product cost = manufacturing cost + general expenses | **65,316MM$** |
| Gross earnings cost = total income—total product costamount of gross earnings cost depends on amount of gross earnings for entire company and income—tax regulations; a general range for gross—earnings cost is 30% to 40% of gross earnings) | **38,428MM$** |
| Profit before Taxes = Revenue–Total Production Cost | **20,833MM$** |
| Profit after tax = 0.52 Profit before Taxes | **9406MM$** |
| Cash Flow =Profit after Taxes + Depreciation | **19,423MM$** |

## 4. Conclusions

The HDO thermodynamic simulation of lignocellulosic biomass was investigated towards the production of chemical compounds and fuel with a higher added value than the raw material. The thermodynamic data were used to evaluate how the products yield varied with the operating condition (temperature, pressure an $H_2$/bio-oil molar ratio). Furthermore, the kinetic data were used to

design, then estimate, the reactor units and relative utility cost. Finally, the separation system cost was added to calculate the TCI, equal to 55,271 MM$, and the *EP* equal to 38,234 M$/y. The HDO process was designed to convert 215,000 barrels/day into fuels with an MFSP of 0.406 to 1.465 $/kg (53.9% higher than fossil fuels derived) and chemicals of high added value, such as sorbitol, propanediol, butanediol, ethanol, etc., at the actual market price. The process guaranteed a ROI of 69.18%, with pay of time of 2.48 and DCFROR for a plant life of 30 years was rating 19.11%.

**Author Contributions:** G.B. developed the theory and performed the calculations. All authors discussed the results and contributed to the final manuscript.

**Funding:** This research received no external funding.

**Conflicts of Interest:** The authors declare no conflict of interest.

## Nomenclature

| | |
|---|---|
| DCFROR | Discounted cash flow rate of return |
| EP | Economic potential |
| FCI | Fixed-capital investment |
| GGE | Gallon gasoline-equivalent |
| GHG | Greenhouse gases |
| HDO | Hydrodeoxygenation |
| HHV | Higher heating value |
| IBO | Insoluble bio-oil |
| IPV | investment value |
| Keq | Equilibrium constant |
| $K_j$ | Equilibrium constant for j reaction |
| LHSV | Liquid hourly space velocity |
| M$ | Billion dollar |
| MM$ | Million dollar |
| MR | Molar ratio |
| M&S | Marshall and shift index |
| $\Delta H$ | Enthalpy variation |
| $R$ | Ideal constant gas |
| ROI | Return of investment |
| $R_1$ | Vapour recycle fraction |
| $R_2$ | Liquid recycle fraction |
| RPV | Return value |
| SMB | Simulated moved bed |
| $T$ | Temperature |
| TBP | Temperature boiling point |
| TCI | Total capital investment |
| TPC | Total product cost |
| WBO | Water-soluble bio-oil |

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
