# Peer review of "Process and Techno-Economic Analysis for Fuel and Chemical Production by Hydrodeoxygenation of Bio-Oil"

_catalysts, doi:10.3390/catal9121021_

Round 1
Reviewer 1 Report
The very interesting and really well done paper of Sanna and Bagnato studied the upgrading of bio-oil adopting the catalytic hydrogenation to give biofuels and chemical intermediates. The study was very careful, and considered the very important aspect of the presence of the two fractions (water and organic fractions). This very important aspect is often neglected in the literature. The authors carried out a very exhaustive economic analysis of the process, considering the characteristic of the starting material and of the operating conditions, obtaining very interesting results.
The work is well described and the literature properly cited, english form correct. In my opinion a very good piece of work.
Author Response
The authors thank the reviewer for his positive opinion.
Reviewer 2 Report
The article entitled „Process and techno-economic analysis for fuel and chemical production by hydrodeoxygenation of bio-oil“ presents wide range of different evaluations of processes still stay in the same range of issues. Authors underline global problems connected with big consumption of fossil fuels and consequences of this occurrence. Authors present wide spectrum of different results, mixt them and make very deeply evaluation. General aim of their study is evaluate the overall performance of a bio-oil hydrotreating process to transportation fuels and chemicals and explore the appropriate operation variables by the economic criteria.
The article is good prepared but sometimes I felt overwhelmed by the amount of data, comparisons, figures and analyzes presented in this study.
Below please find my comments:
Introduction section is well describe. Authors described many specific examples from other scientist’s publications, putting many important datas, which was helpful to understand the meaning of undertaken research.
L 31, „large amount“ – It is too general for scientific documents, even for Introduction section. Please replace it by specific numerous data, examples need to be put in this place.
L33, please provide specific examples of this problems, it could be important for readers.
L33, using „currently“ and after this giving references from 2010, 2011 even 2015 (ref. 1-3) it is a little bit clumsy. I suggest to find more new publication and citation it in this place. In my opinion it should be no older than 2018.
I found many typing mistakes, please check the text very carefully once again. There are many places without finger space between letters and references (L51: aromatics[4]; L64 et al.[6]; L82 constant[10] and many, many more or °C with underline for example lines: 54, 87, 145 and more; unnessesary gaps in lines 182, 306.
In my opinion all titles of figures and tables are too laconic/concise. I suggest to change the titles to more „describing“. Please also provide the information which fig./tab. was taken from references and which ones were made by the authors.
Please replace the year of publication by appropriate number of references in brackets [], lines 103, 106.
I expect figures 2 and 3 between lines 128 and 129 becouse this two figures was mentioned in line 123.
I suggest to use „Methods“ and „Conclusions“ in place of „Method“ and „Conclusion“.
L1, there is tittle missing and full name of jurnal.
Author Response
The authors are grateful about the comments received.
The sentence in L31 has been changed in
“In 2018, fossil fuels’ share in global energy production was 136,580 TWh (93.6 % of the total) [1] contributing ”
[1] BP. Statistical Review of World Energy. 2019; Available from: https://www.bp.com/en/global/corporate/energy-economics/statistical-review-of-world-energy.html.
L33 The following references were added:
Pfleiderer, P., et al., Summer weather becomes more persistent in a 2 °C world. Nature Climate Change, 2019. 9(9): p. 666-671. Blöschl, G., et al., Changing climate both increases and decreases European river floods. Nature, 2019. 573(7772): p. 108-111.We prefer not to add detailed information on the impact of climate change to the planet and human life, since they are out of scope here and it is of general knowledge.
Furthermore, the previous references (1-3) were replaced and renumbered with:
Strezov, V. and H.M. Anawar, Renewable Energy Systems from Biomass: Efficiency, Innovation and Sustainability. 2018: CRC Press. Lindfors, C., et al., Standard liquid fuel for industrial boilers from used wood. Biomass and Bioenergy, 2019. 127: p. 105265. Krutof, A. and K.A. Hawboldt, Upgrading of biomass sourced pyrolysis oil review: focus on co-pyrolysis and vapour upgrading during pyrolysis. Biomass Conversion and Biorefinery, 2018. 8(3): p. 775-787.Thank you for spotting the types. All the typing mistake have been corrected.
Tables and figures captions were improved as suggested by the reviewer.
Method and conclusion were modified in methods and conclusions.